# Differentiating Lung Nodules Due to *Coccidioides* from Those Due to Lung Cancer Based on Radiographic Appearance

**DOI:** 10.3390/jof9060641

**Published:** 2023-06-01

**Authors:** Michael W. Peterson, Ratnali Jain, Kurt Hildebrandt, William Keith Carson, Mohamed A. Fayed

**Affiliations:** 1Fresno Department of Medicine, University of California (San Francisco), San Francisco, CA 93701, USA; 2UCSF Fresno/Community Medical Centers’ Multidisciplinary Lung Nodule Clinic, Fresno, CA 93701, USA; 3Community Medical Imaging Radiology Group, Fresno, CA 93721, USA

**Keywords:** lung nodule, cocci nodule, incidental nodule, lung cancer, pulmonary coccidioidomycosis, valley fever, lung cancer screening, chest CT scan

## Abstract

Background: Coccidioidomycosis (cocci) is an endemic fungal disease that can cause asymptomatic or post-symptomatic lung nodules which are visible on chest CT scanning. Lung nodules are common and can represent early lung cancer. Differentiating lung nodules due to cocci from those due to lung cancer can be difficult and lead to invasive and expensive evaluations. Materials and Methods: We identified 302 patients with biopsy-proven cocci or bronchogenic carcinoma seen in our multidisciplinary nodule clinic. Two experienced radiologists who were blinded to the diagnosis read the chest CT scans and identified radiographic characteristics to determine their utility in differentiating lung cancer nodules from those due to cocci. Results: Using univariate analysis, we identified several radiographic findings that differed between lung cancer and cocci infection. We then entered these variables along with age and gender into a multivariate model and found that age, nodule diameter, nodule cavitation, presence of satellite nodules and radiographic presence of chronic lung disease differed significantly between the two diagnoses. Three findings, cavitary nodules, satellite nodules and chronic lung disease, have sufficient discrimination to potentially be useful in clinical decision-making. Conclusions: Careful evaluation of the three obtained radiographic findings can significantly improve our ability to differentiate benign coccidioidomycosis infection from lung cancer in an endemic region for the fungal disease. Using these data may significantly reduce the cost and risk associated with distinguishing the cause of lung nodules in these patients by preventing unnecessary invasive studies.

## 1. Introduction

Coccidioidomycosis (cocci) is an endemic fungal disease present in California, Arizona and parts of Mexico and South America. Like other endemic fungi, the primary route of infection is inhalation of spores into the lungs. While 60% of cocci infections may be asymptomatic, both asymptomatic and symptomatic pulmonary infections can result in radiographically visible lung nodules [1]. Lung nodules are common findings on chest CT scans and are increasing in frequency [2]. Lung nodules often present a diagnostic dilemma to clinicians [3]. While lung nodules are most often benign, they can represent early lung cancer, and it is important to diagnose lung cancer at an early stage when it is potentially curable.

Because it is important to diagnose malignant disease early but not subject patients to unnecessary and expensive testing, guidelines have been developed to assist clinicians in directing further evaluations [4,5,6,7]. The guidelines start with assigning risk based on clinical parameters that include exposures including tobacco, presence of coexisting pulmonary disease, and family and personal history for cancer. In addition, risk is determined, in part, by the radiographic appearance of the nodule. The primary risk nodule is the one with 8 mm diameter; risk for lung cancer increases with nodule diameter while the risk is low with nodules less in diameter. Risk calculators have been developed and made available on the internet to assist physicians in planning subsequent evaluations [8]. Each of them calculates a percentage risk for the nodule representing lung cancer and guiding additional testing [3].

The potential limitation with these guidelines is that they were not developed or rigorously tested in regions of the country endemic for cocci and have not been tested in patient populations at risk for cocci infection. Guidelines generally suggest CT surveillance for nodules <8 mm in diameter and consideration of biopsy or FDG PET/CT scanning for nodules ≥8 mm [3,9]. However, lung nodules due to cocci are frequently larger than 8 mm in diameter [10] and PET scanning can be falsely positive in both histoplasmosis and cocci [11,12]. In addition, usual evaluations for lung nodules in endemic regions have lower specificity for lung cancer than in non-endemic regions [11,13]. This problem will continue to increase as the rate of lung nodules seen on incidental CT scans and screening CT scans increase [2,14] and can lead to significant costs for the patients and the healthcare systems [15]. Thus, the clinician is faced with a difficult decision when presented with lung nodules ≥ 8 mm in diameter in areas endemic for cocci.

In order to facilitate the evaluation of lung nodules and improve the early diagnosis of lung cancer, we developed a multidisciplinary lung nodule clinic in Fresno, California. Fresno is in the Central Valley of California, an endemic area for cocci. Since cocci can present as lung nodules, we anticipated the need for better diagnostic tools and prospectively collected demographic, clinical and radiological data on our patients as they were seen. Up to one-third of our patients sent to biopsy procedures because of concern for lung cancer were ultimately diagnosed with cocci. This rate is similar to prior studies conducted in endemic fungal regions [10,16].

Because specific diagnosis of lung nodules often requires invasive and expensive testing, we are interested in developing tools to improve our non- or minimally invasive tests to effectively influence post-test probability. While there is much data on the radiographic characteristics of malignant nodules [3,17], nodules due to cocci share many of these radiographic characteristics [10]. Four studies have developed composite prediction models that can be implemented in clinical practice using demographic and radiological data [7]. However, none of those studies were conducted in a population from an endemic area of fungal infections. Thus, we have little data for differentiating lung nodules due to lung cancer from those due to cocci using radiographic imaging. For that reason, we performed this study using our lung nodule patient database to determine what, if any, radiographic characteristics could help differentiate malignant from infectious nodules without the need for early biopsy.

## 2. Materials and Methods

### 2.1. Patient Population

The Multidisciplinary Lung Nodule Clinic is located at the Community Regional Medical Center, a community-based, UCSF Fresno-affiliated academic teaching hospital in Fresno, California. Any patient with a lung nodule on chest imaging can be referred to the clinic regardless of insurance status. These patients are initially evaluated by a pulmonary specialist in the clinic and usual demographic data related to lung cancer risk is collected and documented. These include tobacco smoking history, history of lung disease, personal history of cancer, family history of lung cancer in first degree relatives, and occupational history including history of asbestos exposure. These data are maintained in a patient registry (IRB #2019020). Once evaluated, the patients are reviewed by a multispecialty team who determines further diagnostic testing or treatment. This team include pulmonologists, thoracic surgeons, radiologists, pathologists, radiation oncologists and medical oncologists. The vast majority of patients seen in the clinic reside in a 4-county area of the Central San Joaquin Valley which is endemic for cocci.

### 2.2. CT Interpretation

All patients examined in the clinic had chest CT scans performed for clinical purposes. The study population included patients with a pathologic diagnosis of lung cancer, proven cocci (presence of cocci spherules on biopsy or positive culture for cocci) or probable cocci (granulomatous inflammation without another known cause and positive serologic tests for cocci). The serological testing used during this study were immunodiffusion and complement fixation. We selected scans performed on the nearest date to the clinical diagnosis and provided them to two experienced chest radiologists (KH and WKC) who were blinded to the final diagnosis. Prior to the CT scan review, we developed a scoring sheet of variables that could be easily assessed and that may assist in differentiating the cause of the lung nodules. These included nodule diameter in greatest dimension, anatomic nodule location, nodule density (solid, mixed or ground glass), border characteristics (smooth, lobulated or spiculated), presence of calcification, presence of cavitation including the cavity wall thickness, presence of satellite nodules (1–2 mm in diameter within 3 mm of the primary nodule), multiple or solitary nodules, presence of radiographic evidence for chronic lung disease (emphysema, honey combing, reticular changes or bronchiectasis) and presence of mediastinal adenopathy (>1 cm diameter). The CT scans performed with 5 mm reconstruction and were then reviewed.

### 2.3. Statistical Analysis

Frequency distributions were prepared and examined on all variables and data were carefully reviewed for statistical outliers. Cross tabulations were completed to assess the difference between cocci and lung cancer patients on the radiographic characteristics listed above. To assess associations among categorical variables, we used Pearson’s Chi-Square test, and we used Student’s t test to compare means of different groups. Finally, using the three non-continuous variables that were identified in the multivariate model (satellite nodules, cavitary nodules and chronic lung disease), we calculated their diagnostic utility using sensitivity and specificity. All statistical analyses were performed using IBM/SPSS software, version 25.0 (SSPS, Chicago, IL, USA). Two-sided tests were used, and a *p* value < 0.05 was considered statistically significant.

## 3. Results

We obtained 1079 individual sequential patient records from the Lung Nodule Program. After evaluation by the multidisciplinary team, 474 of these subjects were selected to undergo invasive diagnostic studies. Of these 474 subjects, 192 were diagnosed with bronchogenic carcinoma, 110 were diagnosed with cocci, and 9 had another specific pathological diagnosis. A total of 63 subjects had nonspecific findings on biopsy.

Of the 110 patients with cocci, the average age (±S.D.) was 51.6 (±13.6) years and of the 192 with bronchogenic carcinoma, the average age (±S.D.) was 67.9 (±11.6) years. Among the patients with cocci, 65.5% were male and among the patients with bronchogenic carcinoma, 48.4% were male. Table 1 lists the results of the analysis of the radiographic appearance of the nodules comparing patients with cocci to those with lung cancer. While nodule diameter was greater in patients with lung cancer, the mean diameter for patients with cocci was greater than 2 cm, a nodule diameter considered high risk for lung cancer. Several radiographic findings were indistinguishable between the two diagnoses demonstrating the challenge in differentiating the diseases on radiographic appearance. However, some of the radiographic characteristics did separate the two diseases during the univariate analysis. Cocci nodules more commonly had smooth borders while lung cancer nodules were more commonly spiculated; cocci nodules were more commonly cavitary and had satellite nodules while nodules due to lung cancer were more common in patients with radiographic evidence of chronic lung disease. See Figure 1A,B, and Figure 2A,B.

We next applied multivariate analysis to the differences that were evident after univariate analysis and included age and gender in the model. Table 2 demonstrates those differences found after multivariate analysis and includes the odds ratio that the nodule was due to cocci. As can be seen, age and four radiographic findings differed between lung cancer and cocci after this analysis. Nodule diameter, the presence of a cavity, satellite nodules and evidence of chronic lung disease were all significantly different between the two diagnoses. Interestingly, the cavity wall thickness did not differ between the two disease unlike what has previously been reported on chest X-ray [18].

While nodule diameter and patient age differed between the two diagnoses, they are continuous variables and were not significantly discriminatory. With regard to age, while the patients with lung cancer were older than those with cocci, more than half of them were over the minimum age to qualify for lung cancer screening [9]. The same is true of the nodule diameter in patients with cocci. The average nodule diameter is >2 cm, a size that puts a nodule at high risk for cancer. However, the three non-continuous variables, cavitary nodules, satellite nodules and presence of chronic lung disease, could provide decision support for the clinician. To determine their capacity to direct clinical reasoning, we tested them for sensitivity and specificity. As shown in Table 3, satellite nodules and absence of chronic lung disease had reasonable sensitivity and specificity to predict cocci as the cause of the nodule. While a cavitary nodule was very specific, it lacked sensitivity.

## 4. Discussion

Lung nodules are common on chest CT scans and are increasing in frequency [2]. In addition, as low-dose chest CT scanning for lung cancer increases, clinicians will be seeing more patients presenting for evaluations of lung nodules [19]. Since a proportion of these nodules are early and potentially curable lung cancers, it is important that these cancers be diagnosed in a timely fashion. For those nodules that represent non-malignant disease, it is equally important that we avoid unnecessary testing and invasive studies. This is a decision that challenges the clinical decision-making capability of even experienced physicians.

Fungal infections including histoplasmosis and coccidioidomycosis can result in lung nodules that can be difficult to distinguish from lung cancer [10]. Since up to 60% of primary cocci infections are asymptomatic or minimally symptomatic but can still result in lung nodules, these patients can be particularly challenging. This challenge is evident in cases seen in areas endemic where almost one-third to one-half of high-risk radiographic lung nodules are due to cocci [16]. Several risk calculators have been developed, tested and are available online to assist physicians in calculating risk for lung cancer in lung nodules [7]. However, these calculators were not developed or tested in geographic regions with endemic fungal disease. When we applied these calculators to our patients, they had very poor specificity and positive predictive value in patients with cocci and frequently identified cocci patients as moderate or high risk for lung cancer. Similarly, PET scans are recommended to evaluate moderate risk lung nodules [3]. However, PET scans can be positive in cases of lung nodules due to cocci [11], and PET scans of lung nodules performed in endemic fungal regions have a lower specificity than those performed in non-endemic regions [13].

Mischaracterizing nodules due to cocci as intermediate to high risk for lung cancer potentially exposes the patient to risks associated with invasive procedures and unnecessary costs. In addition, patients can experience significant anxiety inherent in a potential cancer diagnosis. Since many of nodules due to cocci and to lung cancer present as peripheral nodules, they are frequently biopsied by CT-guided transthoracic needle biopsy [20]. In our own institution, transthoracic needle biopsy is associated with a 25% incidence of complications [21]. Finally, we estimated that the annual cost of cocci in California is USD 700 million, and the most costly forms of the disease are disseminated disease (occurring in 1% of infections) and lung nodules requiring invasive studies [15].

These data show that it can be challenging to differentiate nodules due to cocci from those due to lung cancer. However, to our knowledge, this is the first study that objectively collected and analyzed the data on radiographic imaging between these two diseases. Our study demonstrates that some radiographic findings together with age can potentially improve the predictive value of imaging performed in patients who may have been exposed to cocci. Unfortunately, while our study has identified findings that will influence the risk, none of them are sufficiently discriminating to be used alone to drive the decision. Future studies will need to include more patients and include both clinical and radiographic characteristics to improve our predictive models. We are currently working on such a model.

While cocci has a limited geographic range in southwestern United States and the Central Valley of California, more than 13 million people are potentially exposed and are at risk for the infection within the existing boundaries. In addition, the disease is increasing in this endemic region [22], and appears to be geographically expanding beyond its former boundaries [23,24]. A recent study using Medicare database data found that cocci occurred above the clinically relevant threshold in one or more counties in 35 states and the district of Columbia (69%). Cocci climate modeling provides more data which suggest that the disease will expand even further over the next 50 years [25]. Thus, more clinicians will see lung nodule patients who have travelled through or lived in cocci endemic regions and be faced with this difficult clinical decision.

The strength of this study is that it used a real-world population that had lung nodules found both incidentally and when conducting imaging to evaluate respiratory complaints. In addition, our practice is in an academic affiliated community hospital and not a tertiary level referral center. Patients were also examined in a multidisciplinary clinic where multiple specialists discussed all the patients’ conditions.

Two weaknesses of this study are that it was conducted at a single institution and the chest CT scans were retrospectively reviewed by chest radiologists. We look forward to other centers reporting their observations to compare with ours and to studying our own patient population based on clinical radiographic interpretations in a clinical setting. In addition, our sample did not include patients identified using low dose CT screening so our results may not reflect a different and higher risk population with a different pre-test probability for lung cancer.

## Figures and Tables

**Figure 1 jof-09-00641-f001:**
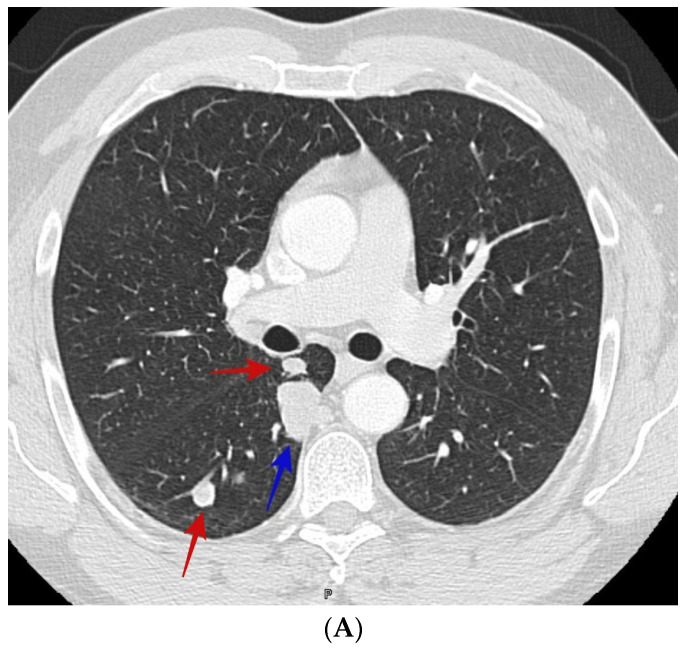
(**A**) CT scan shows smooth right upper lobe lung nodule (blue arrow) and satellite nodules (red arrows). The patient was a 50-year-old male who presented with incidental lung nodules. CT guided biopsy showed coccidioidomycosis. (**B**) shows sagittal section of the right upper lobe nodule (blue arrow) and satellite nodules (red arrows) of the same patient in image Figure 1A.

**Figure 2 jof-09-00641-f002:**
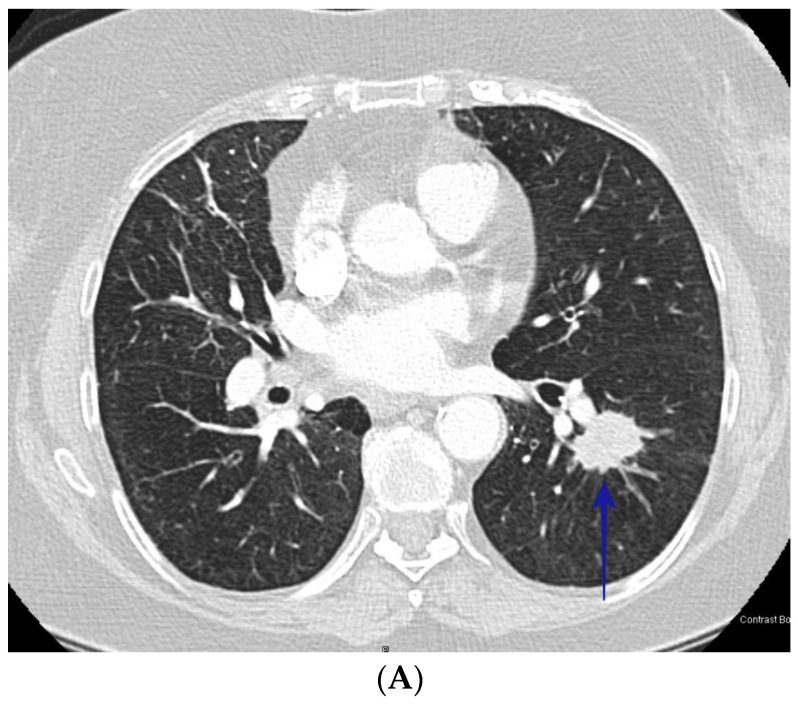
(**A**) CT scan shows speculated left lower nodule (blue arrow). The patient was a 76-year-old female who presented with lung nodules. CT guided biopsy showed squamous cell carcinoma. (**B**) shows sagittal section of the left lower lobe nodule (blue arrow) of the same patient in image Figure 2A.

**Table 1 jof-09-00641-t001:** Univariate analysis of radiographic findings.

Radiographic Characteristics	Coccidioidomycosis	Lung Cancer	*p* Value
Nodule Diameter (cm)	2.59 ± 1.7	4.08 ± 2.5	<0.001
Present in upper lobe	46.5%	64.7%	0.003
Present in the lower lobe	53.5%	35.3%	0.003
Solid Density	80.7%	84%	NS
Border Character			
Smooth	30.9%	5.2%	<0.001
Lobulated	31.8%	38.5%	NS
Spiculated	32.7%	56.3%	<0.001
Calcification	4.6%	2.7%	NS
Cavitary	24.8%	8.8%	<0.001
Cavity Wall Thickness (mm)	4.8 ± 2.7	4.4 ± 2.9	NS
Satellite Nodules Present	58.7%	14.3%	<0.001
Multiple nodules	71.6%	58.8%	0.02
Chronic Lung Disease	18.3%	66%	<0.001
Mediastinal Adenopathy	56.9%	60.3%	NS

**Table 2 jof-09-00641-t002:** Multivariate analysis of radiographic findings, adjusted for age and gender.

Variables	Odds Ratio for Coccidioidomycosis	95% CI	*p* Value
Nodule Diameter (cm)	0.514	0.383–0.688	<0.001
Present in upper lobe	0.660	0.278–1.567	0.346
Border Character			
Smooth	3.380	0.939–12.173	0.062
Spiculated	1.222	0.481–3.105	0.673
Cavitary	7.062	1.757–28.380	0.006
Satellite Nodules Present	17.781	5.644–56.016	<0.001
Multiple nodules	1.058	0.381–2.936	0.913
Chronic Lung Disease	0.192	0.075–0.492	0.001
Age	0.927	0.896–0.959	<0.001
Gender	0.486	0.205–1.154	0.102

**Table 3 jof-09-00641-t003:** Sensitivity and specificity for three radiographic findings for coccidioidomycosis.

Radiographic Finding	Satellite Nodules	Cavitary Nodule	Absence of Chronic Lung Disease
Sensitivity (95% CI)	70.2% (70.8–86)	24.6% (16.8–33.7)	66.1% (59–72.8)
Specificity (95% CI)	86% (80.2–90.5)	91.2% (86.2–94.8)	81.8% (73.3–88.5)
Positive Likelihood Ratio (95% CI)	5.63 (3.9–8.1)	2.77 (1.6–4.9)	3.64 (2.42–5.48)
Negative Likelihood Ratio (95% CI)	0.24 (0.17–0.35)	0.83 (0.74–0.93)	0.41 (0.33–0.51)
Positive Predictive Value (95% CI)	70.7% (62.7–77.6)	54.3% (40.4–67.5)	60.8% (50.9–70.1)
Negative Predictive Value (95% CI)	90.6% (87.1–93.2)	73.8% (71.5–76)	84.9% (82–87.5)
Accuracy	83.9% (79.4–87.8)	71.2% (65.7–76.2)	77.1 (72–81.7)

## Data Availability

The data presented in this study are available on request from the corresponding author. The data are not publicly available due to privacy restriction.

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
