# Peer review of "Differentiating Lung Nodules Due to Coccidioides from Those Due to Lung Cancer Based on Radiographic Appearance"

_jof, 2023, doi:10.3390/jof9060641_

Round 1
Reviewer 1 Report
I consider that is necessary to mention in the text which serology test was used for the diagnosis of coccidoidomycosis, since the specificity and sensitivity of these tests differ.
Author Response
Reviewer 1
Comments and Suggestions for Authors
I consider that is necessary to mention in the text which serology test was used for the diagnosis of coccidioidomycosis, since the specificity and sensitivity of these tests differ.
Answer: I added all tests performed in the study. In this study 2 methods were used immunodiffusion and complement fixation
Reviewer 2 Report
The authors present a brief study comparing the pulmonary nodules and associated findings on chest CT scans in coccidioidomycosis and lung cancer. The results of the study are relevant for the population where coccidioidomycosis has moderately high prevalence that it is an important differential, where investigations of a pulmonary nodule may not prove cost-effective in preventing cancer.
It is a simple well-presented purely radiological data comparison between the 2 diseases. The authors may have added a couple of images to make the dilemma more visually apparent. The authors may consider adding coccidioides serology along with clinical and radiological data in future studies to develop better predictive models to rule out cocci in endemic regions.
Author Response
The authors present a brief study comparing the pulmonary nodules and associated findings on chest CT scans in coccidioidomycosis and lung cancer. The results of the study are relevant for the population where coccidioidomycosis has moderately high prevalence that it is an important differential, where investigations of a pulmonary nodule may not prove cost-effective in preventing cancer.
It is a simple well-presented purely radiological data comparison between the 2 diseases. The authors may have added a couple of images to make the dilemma more visually apparent. The authors may consider adding coccidioides serology along with clinical and radiological data in future studies to develop better predictive models to rule out cocci in endemic regions.
Answer: It is ongoing. thanks
Reviewer 3 Report
The paper by Peterson et al. suggests that radiographic findings of chest CT images can be used to distinguish lung cancer from Coccidioidomycoses (cocci) in Coccidioides endemic areas. This is an interesting problem as lung nodules can be seen in both and it would be beneficial to patients to have a non-invasive way of distinguishing the diseases so that early treatment can be started for either disease state. A regional hospital associated with an academic medical center in Fresno California was used to retrospectively score CT images of patients with lung nodules and suspected lung cancer for several parameters and then perform univariate and then multivariate analysis to determine risk for lung cancer of cocci. Specific comments include:
1. There is very little patient data and since the size of the nodules was included, it would be useful to know the time from onset of symptoms that the CT scans were performed. Otherwise, the size of the nodule isn't really that useful.
2. The scoring rubric for the radiographic findings are not included in the paper.
3. It would be useful for non-clinicians to include sample CT images pointing out the parameters being analyzed.
4. The authors indicate they are performing a larger study with more patients from other medical centers. However, it would be interesting to know whether other radiologist could use the scoring rubric and predict diagnosis of retrospective images or even the same images used to do the analysis in order to validate this is a useful tool.
Author Response
The paper by Peterson et al. suggests that radiographic findings of chest CT images can be used to distinguish lung cancer from Coccidioidomycoses (cocci) in Coccidioides endemic areas. This is an interesting problem as lung nodules can be seen in both and it would be beneficial to patients to have a non-invasive way of distinguishing the diseases so that early treatment can be started for either disease state. A regional hospital associated with an academic medical center in Fresno California was used to retrospectively score CT images of patients with lung nodules and suspected lung cancer for several parameters and then perform univariate and then multivariate analysis to determine risk for lung cancer of cocci. Specific comments include:
- There is very little patient data and since the size of the nodules was included, it would be useful to know the time from onset of symptoms that the CT scans were performed. Otherwise, the size of the nodule isn't really that useful.
Answer: most of the patients had incidental abnormal chest imaging and no significant symptoms.
- The scoring rubric for the radiographic findings are not included in the paper.
Answer: I included examples of the imaging
- It would be useful for non-clinicians to include sample CT images pointing out the parameters being analyzed.
Answer: I included examples of the imaging
- The authors indicate they are performing a larger study with more patients from other medical centers. However, it would be interesting to know whether other radiologist could use the scoring rubric and predict diagnosis of retrospective images or even the same images used to do the analysis in order to validate this is a useful tool.
Answer: I definitely agree and therefore next step is to look at all pathological diagnosis of coccidioidomycosis and malignancy, then perform retrospective review of the images.
Reviewer 4 Report
The authors contributed the work entitled “Differentiating Lung Nodules due to Coccidioides from those due to Lung Cancer by Radiographic Appearance".
This review covers the knowledge in the field and expands the current understanding of the field. Work is quite interesting, and the reviewer recommended its publication after clarifying of the following:
-What is the difference between this work and the work previously published by the authors? “Differentiating incidental lung nodules due to lung cancer from those due to coccidioidomycosis”. European Respiratory Journal 2015 46: PA2996; DOI: 10.1183/13993003.congress-2015.PA2996
Author Response
The authors contributed the work entitled “Differentiating Lung Nodules due to Coccidioides from those due to Lung Cancer by Radiographic Appearance".
This review covers the knowledge in the field and expands the current understanding of the field. Work is quite interesting, and the reviewer recommended its publication after clarifying of the following:
-What is the difference between this work and the work previously published by the authors? “Differentiating incidental lung nodules due to lung cancer from those due to coccidioidomycosis”. European Respiratory Journal 2015 46: PA2996; DOI: 10.1183/13993003.congress-2015.PA2996
Answer: it is the same data base that were used to determine the accuracy of the existing lung cancer calculators.
Round 2
Reviewer 4 Report
_